# Twilight: Adaptive Attention Sparsity with Hierarchical Top-$p$ Pruning

**Chaofan Lin    Jiaming Tang    Shuo Yang    Hanshuo Wang**
**Tian Tang    Boyu Tian    Ion Stoica    Song Han    Mingyu Gao**
Tsinghua University    Massachusetts Institute of Technology
University of California, Berkeley
`lcf24@mails.tsinghua.edu.cn   gaomy@tsinghua.edu.cn`

`https://github.com/tsinghua-ideal/Twilight`

## Abstract

Leveraging attention sparsity to accelerate long-context large language models (LLMs) has been of great importance recently. However, most existing sparse attention algorithms use a fixed budget of how many tokens to use in their computations. This simple static decision raises critical issues in real-world deployment because it fails to account for the dynamic nature of real-world scenarios, where the optimal balance between accuracy and efficiency can vary greatly. In this paper, we reveal a key insight that leveraging the idea of top-$p$ sampling (a.k.a., nucleus sampling) in sparse attention could enable efficient and adaptive budget decisions. Based on this, we propose Twilight, a framework that enhances any existing sparse attention algorithm with adaptive budget decision capabilities without sacrificing accuracy. Empirical results show that Twilight can adaptively prune up to 98% tokens with nearly no accuracy loss in both long- and medium-context scenarios, leading to a $1.4\times$ speedup over state-of-the-art sparse attention mechanisms.

## 1 Introduction

Large language models (LLMs) with long-context capabilities have revolutionized a wide array of natural language processing applications, such as retrieval-based tasks, document summarization [1], and code generation [16]. The increasing availability of models supporting context windows up to 1M to 10M tokens [47, 37] highlights the growing potential of these advancements. For instance, video language models (VLMs) [41] often require tens of thousands of tokens for video processing. Similarly, large reasoning models [8, 38], which are rapidly growing in popularity, frequently demand substantial token lengths to enable chain-of-thought (CoT) reasoning. Consequently, the importance of long-context LLMs is increasing rapidly to meet the needs of these sophisticated applications.

Despite the substantial potential of long-context LLMs, they come with excessive computational and memory costs [58, 36, 45], primarily from the attention mechanism. Particularly, in the decoding stage of LLMs, the key-value (KV) cache size grows rapidly as the token sequence becomes longer. These data need to be repeatedly loaded from the memory, leading to significant latency overheads. Furthermore, the substantial size of the KV cache significantly increases the GPU memory consumption, compounding the challenges of continuously scaling long-context LLMs.

Previous research has extensively investigated the use of *attention sparsity* (a.k.a., KV cache sparsity) to accelerate long-context inference, both during the prefilling and decoding stages. The core idea is to compute an approximate attention on a subset of tokens, often referred to as "critical tokens" or "heavy hitters" [58]. The number of selected tokens, denoted as $B$, is commonly referred to as the KV cache *budget*. A top-$k$ operation is required to identify the indices of the critical tokens that

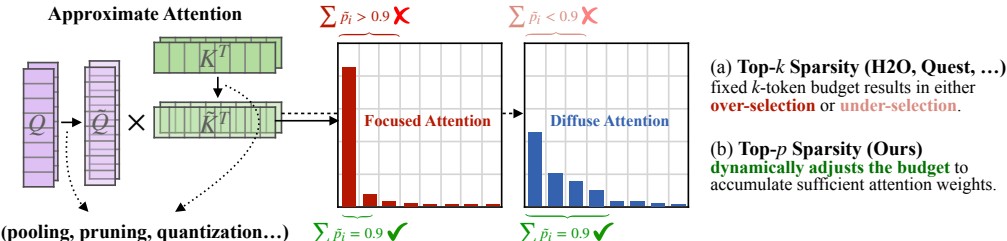

**Approximate Attention**

$\sum \tilde{p}_i > 0.9$ ✘    $\sum \tilde{p}_i < 0.9$ ✘

**Focused Attention**    **Diffuse Attention**

(a) **Top-$k$ Sparsity (H2O, Quest, …)** fixed $k$-token budget results in either **over-selection** or under-selection.

(b) **Top-$p$ Sparsity (Ours)** **dynamically adjusts the budget** to accumulate sufficient attention weights.

(pooling, pruning, quantization…)    $\sum \tilde{p}_i = 0.9$ ✔    $\sum \tilde{p}_i = 0.9$ ✔

Figure 1: Comparison of top-$k$ and top-$p$ for attention sparsity. Approximate attention typically employs techniques such as pooling, channel pruning, and quantization to approximate the query ($\tilde{Q}$) and key ($\tilde{K}$) and estimate the attention weights. These weights are then used to select important tokens for sparse attention. (a) Top-$k$ sparsity, utilized by most existing designs, relies on a fixed $k$-token budget and often results in **over-selection** ($\sum \tilde{p}_i > 0.9$) or **under-selection** ($\sum \tilde{p}_i < 0.9$). (b) Our proposed top-$p$ sparsity **dynamically adjusts the budget** to accumulate just sufficient attention weights ($\sum \tilde{p}_i = 0.9$), enabling more efficient and adaptive sparse attention.

correspond to the $B$ highest estimated scores. However, a key tradeoff exists for the selection of the budget. A smaller $B$ value significantly reduces the memory accesses and computations, while a larger $B$ value retains more contextual information and thereby minimizes the accuracy loss.

Identifying the optimal value of $B$ to balance both accuracy and efficiency is inherently challenging due to two major reasons: **(a) The best budget choices vary dynamically at runtime.** Previous works [44, 45] have demonstrated that some heads, referred to as "retrieval heads", are trained to extract important information from long contexts, while others focus only on local information. From Figure 1 we see that the distribution of attention weights may vary across different attention heads. Some attention distributions concentrate on a small subset of tokens, which we refer to as *focused attention*. Other attention distributions may be flatter, where many tokens have similar attention weights; we call this *diffuse attention*. For focused attention, using a

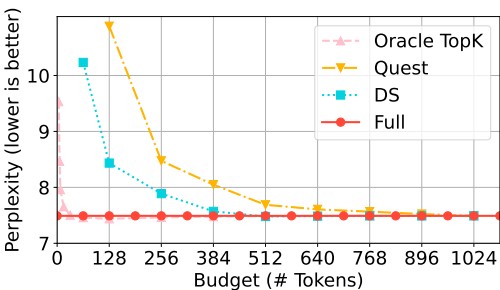

Figure 2: Relationship between the KV cache budget and the perplexity on the PG-19 dataset in different top-$k$ sparse attention methods.

fixed token budget for top-$k$ attention often leads to over-selection, as only a few tokens are needed to accumulate sufficient attention weights. In contrast, for diffuse attention, a fixed budget can result in under-selection, as a larger number of tokens are necessary to ensure accurate attention modeling. **(b) Existing algorithms need different degrees of over-selection to compensate the estimation inaccuracy.** As shown in Figure 2, the optimized budgets highly depend on the specific algorithms, necessitating offline calibration to determine the appropriate budget for each algorithm individually. Certain methods, like Quest [36] or DS [50], have to over-select some tokens to compensate for the inevitable inaccuracy in estimating the importance of tokens compared to the oracle.

In this work, we reveal that the top-$k$ methods exhibit issues similar to those previously encountered in LLM sampling. Drawing on this analogy, we introduce top-$p$ sampling into sparse attention to address the budget selection dilemma. Our study demonstrates that *top-$p$ can determine the KV cache budget in a more intrinsic and dynamic way compared to top-$k$*. Based on these observations, we build Twilight, a hierarchical KV cache pruning framework that enhances existing sparse attention algorithms with *adaptive budget selection* capabilities. Specifically, Twilight first lets the base algorithm select a relatively large subset of tokens using a conservative budget, and then further refines this subset by retaining only the top-$p$ tokens.

Our evaluations for Twilight are conducted in two aspects: accuracy and efficiency. First, we demonstrate that Twilight optimizes the base algorithms with nearly no accuracy loss on both medium-context benchmarks (GSM8K [4], COQA [33], PG-19 [32]) and long-context benchmarks (Longbench [1], RULER [14]). Next, we show that Twilight achieves up to $15.8\times$ performance

improvement over the full attention operation. Compared to prior sparse attention methods, Twilight enables a $1.4\times$ speedup for the self-attention operator itself, and a $1.35\times$ speedup for the end-to-end decoding. Our contributions are summarized as follows:

- We conduct an in-depth investigation into a critical challenge in top-$k$ sparse attention: the difficulty in identifying the optimal budget (i.e., the number of selected tokens). We propose to use top-$p$ sampling instead to dynamically determine this budget at runtime.

- We introduce Twilight, a framework that can endow any existing sparse attention method with adaptive budget selection capabilities, thereby further improving their efficiency.

- We evaluate Twilight in terms of both accuracy and efficiency, demonstrating a speedup of $1.4\times$ over existing sparse attention methods with nearly no accuracy loss.

## 2 Related Work

**Top-$k$ Sparse Attention.** H2O [58], StreamingLLM [46], and SnapKV [23] evict non-critical tokens in static, query-agnostic manners, which are often referred to as KV cache compression. In contrast, SparQ [34], Quest [36], Double Sparsity (DS) [50], and HShare [43] retain all tokens in the GPU memory but select critical tokens to save data accesses. Recent works like RetrievalAttention [26] and PQCache [55] adopt advanced algorithms to better estimate the token criticality. NSA [54] and MoBA [29] explore opportunities in trainable sparse attention. However, these methods are all based on top-$k$ which requires proper budget selection and configuration beforehand, and thus suffer from the over/under-selection issues.

**Dynamic Budget.** More recent studies have extensively demonstrated that the optimal budgets vary significantly across different layers [2, 48], attention heads [9, 45, 35], and prompts (tasks) [61]. Please see Appendix A for details. These works tend to focus on only one aspect of the dynamism. In this paper, we point out that it is the different distributions of attention weights that are the root cause of such dynamism.

**Non-top-$k$ Sparse Attention.** Some recently emerged designs also go beyond top-$k$ methods. MagicPIG [3] uses locality-sensitive hash (LSH) sampling instead of dropping tokens to estimate attention weights, but requires complicated algorithm-system co-design. SampleAttention [64] also features adaptive sparsity but focuses on the prefill stage. A concurrent work with ours, Tactic [62], also dives into top-$p$ sparsity but it uses function fitting to estimate the weight distributions. Although it potentially has lower estimation cost, it usually overestimates the budget.

**Other KV Cache Optimizations.** Several alternative approaches focus on optimizing the KV cache beyond sparsification, including quantization [13, 27, 17, 30], linear attention [42, 18], and memory-efficient attention mechanisms such as FlashAttention [5] and SageAttention [57, 56]. Our approach is orthogonal to these methods, and can be combined with them for enhanced performance.

## 3 Bringing Top-$p$ Sampling to Sparse Attention

In this section, we formulate the current sparse attention methods and re-examine the root cause of their inefficiencies. We argue that to mathematically approximate the attention, the goal is to select a minimum set of indices such that the sum of their attention scores meets a certain threshold. Therefore, we propose to use top-$p$ sampling instead of top-$k$ to efficiently identify the critical tokens.

### 3.1 Problem Formulation

We start by formulating the sparse attention mechanism. Consider the attention computation during the decoding phase, where we have the query vector $\mathbf{q} \in \mathbb{R}^{1 \times d}$, and the KV cache $\mathbf{K}, \mathbf{V} \in \mathbb{R}^{n \times d}$. Here, $d$ denotes the head dimension, and $n$ represents the context length.

**Definition 3.1** (Sparse Attention). Let $\mathcal{I}$ be the set of selected indices. Sparse attention calculates

$$\hat{\mathbf{o}} = \mathrm{softmax}\left(\frac{\mathbf{q} \cdot \mathbf{K}^T}{\sqrt{d}}\right) \mathbf{\Lambda}_{\mathcal{I}} \mathbf{V} = \mathbf{W} \mathbf{\Lambda}_{\mathcal{I}} \mathbf{V} \in \mathbb{R}^{1 \times d} \tag{1}$$

where $\mathbf{\Lambda}_{\mathcal{I}} \in \mathbb{R}^{n \times n}, \mathbf{\Lambda}_{\mathcal{I}}[i, j] = \begin{cases} 1 & \text{if } i = j \text{ and } i \in \mathcal{I} \\ 0 & \text{otherwise} \end{cases}$.

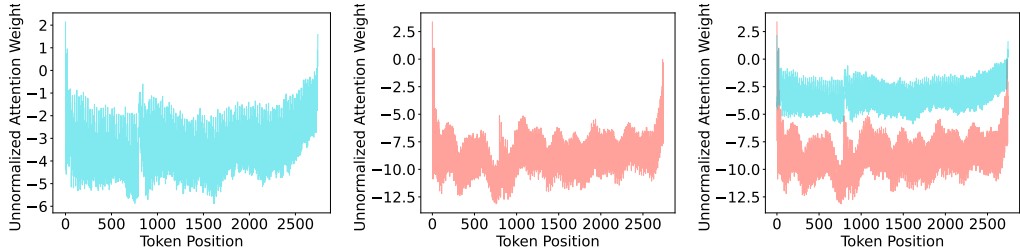

Figure 3: Diverse distributions observed in attention weights. **The leftmost image** illustrates a "flat" distribution (**diffuse attention**), where the weights are close to uniformly distributed. **The middle image** depicts a "peaked" distribution (**focused attention**), where the weights are concentrated on the tokens at the two sides. When overlaid as in **the rightmost image**, the differences between these distributions become readily apparent.

Let the accurate attention output be $\mathbf{o} = \text{softmax}(\frac{\mathbf{q} \cdot \mathbf{K}^T}{\sqrt{d}})\mathbf{V} \in \mathbb{R}^{1 \times d}$. To minimize the error $\|\mathbf{o} - \hat{\mathbf{o}}\|$, we need to carefully select the subset of tokens used in $\mathcal{I}$. However, directly optimizing this objective function without loading the full KV cache is challenging. According to the sub-multiplicative property of the Frobenius norm, we can bound the error as in [Equation 2]. Earlier research has shown that the distribution of $\mathbf{V}$ is relatively smooth [59], which implies $\|\mathbf{V}\|_F$ can be viewed as a constant.

$$\mathcal{L} = \|\mathbf{o} - \hat{\mathbf{o}}\| = \|\mathbf{W}(\mathbf{\Lambda}_{\mathcal{I}} - \mathbf{1}^{n \times n})\mathbf{V}\| \\ \leq \|\mathbf{W}(\mathbf{\Lambda}_{\mathcal{I}} - \mathbf{1}^{n \times n})\| \cdot \|\mathbf{V}\|_F \tag{2}$$

Therefore, the objective becomes minimizing $\|\mathbf{W}(\mathbf{\Lambda}_{\mathcal{I}} - \mathbf{1}_{n \times n})\| = 1 - \sum_{i \in \mathcal{I}} \mathbf{W}[i]$, which means selecting a subset of tokens that maximize the sum of their attention weights. If we fix the size of this subset, i.e. $|\mathcal{I}|$, then we have the oracle top-$k$ attention:

**Definition 3.2** (Oracle Top-$k$ Sparse Attention). Given the budget $B$,

$$\mathcal{I} = \arg\max_{\mathcal{I}} \sum_{i \in \mathcal{I}} \mathbf{W}[i] \quad \text{s.t.} \ |\mathcal{I}| = B \tag{3}$$

This serves as a theoretical upper bound of the current top-$k$ sparse attention methods.

## 3.2 Rethinking the Problem of Top-$k$

The Achilles' heel of top-$k$ sparse attention, as described earlier, is the dilemma in determining a universally applicable budget $B$ to all scenarios. We find that this predicament is quite similar to a previous problem encountered in the sampling phase of LLMs, during which the model samples the final output token from the predicted probability distribution. Nucleus sampling [12], a.k.a., top-$p$ sampling, was proposed to address the problem that top-$k$ sampling cannot adapt to different next-word distributions.

Motivated by this insight, we examine the distributions of attention weights more closely. As indicated by [Equation 2], the output error is bounded by the sum of the selected attention weights. Therefore, the objective should become selecting the minimum number of tokens $B$ to satisfy a given requirement for the output error. [Figure 3] displays two different types of attention weight distributions in several real-world LLMs mentioned in [Figure 1]. It is easy to observe that, compared to the peaked distribution, a greater number of tokens must be selected in the flat distribution to reach the same cumulative threshold.

Therefore, we argue that *the core reason for budget dynamism is the dynamic nature of attention weight distributions at runtime.* We thus introduce top-$p$ sparse attention by directly applying a threshold to the sum of attention weights.

**Definition 3.3** (Oracle Top-$p$ Sparse Attention). Given the threshold $p$,

$$\mathcal{I} = \arg\min_{\mathcal{I}} |\mathcal{I}| \quad \text{s.t.} \ \sum_{i \in \mathcal{I}} \mathbf{W}[i] \geq p \tag{4}$$

Compared to top-$k$, top-$p$ is more advantageous because it provides a theoretical upper bound of error as $(1 - p) \cdot \|\mathbf{V}\|_F$ from Equation 2. Under this circumstance, top-$p$ reduces the budget as low as possible, making it both efficient and adaptive to different distributions. To demonstrate how top-$p$ reduces the budget, we investigate a real distribution of attention scores as shown in Figure 4. Compared to two fixed budget strategies $B = 16$ and $B = 1024$ that respectively result in under-selection and over-selection, the $p = 0.8$ point selects a very small budget $B = 97$ to reach a similar error requirement to that of $B = 1024$.

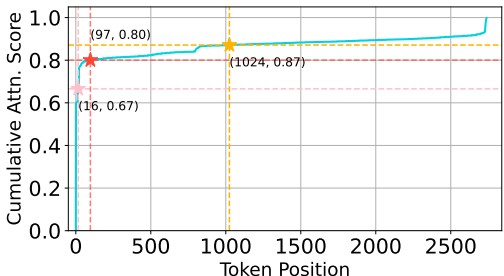

Figure 4: Cumulative attention scores of different budget selections in one example attention head.

## 4 Twilight

With the efficient and adaptive top-$p$ sparse attention, our primary goal is to use it to endow existing algorithms with adaptive budget selection capabilities, rather than simply inventing yet another sparse attention design. We are mainly motivated by two reasons. On one hand, despite the challenge of budget selection, existing sparse attention algorithms have achieved significant success in LLM serving systems [20, 60], thanks to their effective token selection strategies. These strategies can be readily reused and enhanced with our adaptive sparsity. On the other hand, we anticipate that future sparse attention methods may still employ top-$k$ selection. By developing a general solution, we aim to automatically equip these future methods with adaptive attention sparsity, while avoiding extensive redesign. Consequently, we position our system, Twilight, as an **optimizer** for existing algorithms.

Nevertheless, applying top-$p$ to various existing sparse attention algorithms faces three key challenges on both the algorithm and system perspectives. **(C1) Not all algorithms are suitable for top-$p$.** Top-$p$ imposes strict constraints on the layout of attention weights. For example, simply replacing top-$k$ with top-$p$ in

Table 1: Comparison of different pruning methods on attention weights. "Normalization" means `softmax`.

| Method | Efficiency | Precision Requirement | Output Accuracy | Need Normalization? |
|---|---|---|---|---|
| Top-$k$ | High | Low | Median | × |
| Top-$p$ | High | Median | High | √ |
| Full Attn. | Low | High | High | √ |

Quest [36] would not work, as Quest performs max pooling on weights with a per-page layout (16 tokens per page). Additionally, some other methods [49, 26] do not use attention weights to select critical tokens at all. **(C2) It is harder to estimate weights for top-$p$ than top-$k$.** In order to find critical tokens without loading full K data, low-precision representation of the K cache is usually used [50, 55]. However, the precision requirement of top-$p$ is higher than that of top-$k$, because the former requires a certain degree of numerical accuracy while the latter only demands ordinality. Table 1 compares top-$k$, top-$p$, and full attention. The precision requirement of top-$p$ attention lies in between the other two, necessitating reconsideration of appropriate precision choices for the K cache. **(C3) System-level optimizations are needed.** Since our work is the first to introduce top-$p$ to attention weights, the relevant algorithms need to be efficiently implemented on the GPU, including efforts on both parallel algorithm designs and kernel optimizations.

In Section 4.1, we address **C1** by proposing a unified hierarchical pruning framework for top-$p$ sparse attention. In Section 4.2, we mitigate the runtime overheads with efficient kernel implementations (Top-$p$, SpGEMV, Attention) and 4-bit quantization of the K cache, addressing **C2** and **C3**. Lastly, in Section 4.3, we analyze the overheads of Twilight and discuss some additional issues.

### 4.1 Hierarchical Pruning with a Select-then-Prune Architecture

To uniformly support various sparse attention mechanisms, we propose a two-step, hierarchical pruning process. We first capture the base algorithm into a black-box **Token Selector** as long as it has the common semantics of selecting a subset of critical tokens, while the exact algorithm details of *how* to select do not matter. We let the Token Selector use a conservative, relatively large budget,

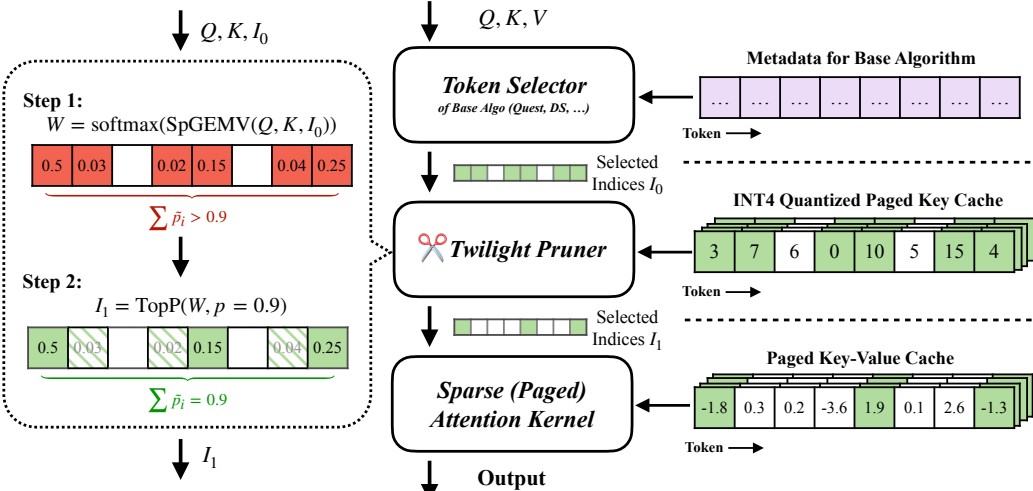

Figure 5: Twilight architecture. Twilight incorporates a certain existing base sparse attention algorithm and further optimizes it. It computes self-attention in three steps. First, the **Token Selector** selects critical tokens using the base algorithm under a conservative budget. Then, the **Twilight Pruner** prunes the selected token subset via top-$p$ thresholding. Finally, the pruned token indices are passed to the **Sparse Attention Kernel** to perform the attention computation.

e.g. $1/4$ sparsity. Then, we have a **Twilight Pruner** after it to further optimize the selected indices by only retaining the top-$p$ tokens, i.e., the minimum subset whose attention weight sum exceeds the threshold $p$. We call this design as the *Select-then-Prune* architecture, as illustrated in the middle of Figure 5. The final sparse attention kernel thus only computes on the top-$p$ tokens, achieving the benefits of efficiency and adaptivity as proved in Section 3.

## 4.2 Efficient Kernel Implementations

Now we briefly describe the details of the Twilight architecture, particularly for the Pruner step. For more details of the kernel implementations, please refer to Appendix B.

**Efficient SpGEMV with 4-bit Quantization of Key Cache.** The beginning part of the Pruner is similar to other sparse attention algorithms, which is to estimate the importance of tokens. As we formulated in Section 3.1, this can be done by estimating the similarity between $\mathbf{q}$ and $\mathbf{K}$, i.e., $\mathbf{q} \cdot \mathbf{K}$. Since loading $\mathbf{K}$ is known to be memory bound, we reduce the memory access cost by quantizing $\mathbf{K}$ into lower precision. But what precision shall we choose? Table 1 shows the precision requirement of top-$p$ lies in between top-$k$ and full attention. Some existing top-$k$ designs [50, 55] have pushed the compression of the K cache to extremely low precisions of 1 to 2 bits. For full attention, SageAttention [57] has demonstrated 8-bit precision with smoothing $\mathbf{K}$ and per-block quantization. In this work, we empirically find that *4-bit precision strikes a balance between accuracy and efficiency for top-p*, as illustrated in Figure 6. Here the sum of attention weights with 2-bit quantization drops significantly, while 4-bit and 8-bit methods both maintain enough stability.

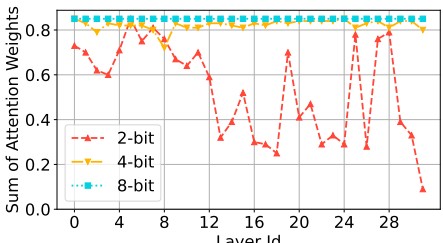

Figure 6: Sums of normalized attention weights for the selected tokens under different quantization precisions, with $p = 0.85$.

Hence we implement an efficient sparse GEMV (SpGEMV) kernel based on FlashInfer [52], a high-performance kernel library for LLM serving. Here "sparse" means the quantized K cache data are stored/loaded in a paged manner [20] to align with the original KV cache layout. We maintain this extra INT4 asymmetrically quantized K cache on the GPU as shown at the right of Figure 5. The INT4 $\mathbf{K}$ vectors are unpacked and dequantized in the shared memory, reducing data accesses from the global memory to at most $1/4$, resulting in considerable end-to-end speedup.

**Efficient Top-$p$ via Binary Search.** A brute-force way to do top-$p$ sampling is to sort the elements by descending order and accumulate them until the sum meets the threshold. This is quite inefficient in parallel hardware like modern GPUs. As our top-$p$ method is motivated by the top-$p$ sampling, we also implement this kernel by modifying the top-$p$ sampling kernel from FlashInfer [52]. Specifically, our kernel adopts a parallel-friendly binary search method as in Algorithm 1. Note that element-wise operations like max, where, and sum can be fused into a single loop, which is tensorized on the GPU. Thus we do not need to materialize the intermediate variables like $W_0$.

---

**Algorithm 1** Top-$p$ via Binary Search.

**Input:** normalized attention weights $W \in \mathbb{R}^{BS \times H \times N}$, top-$p$ threshold $p$, hyper-parameter $\epsilon$.
**Output:** indices $\mathcal{I}$, mask $\mathcal{M} \in \{0, 1\}^{BS \times H \times N}$.

$l = 0, r = \text{max}(W), m = (l + r)/2;$
**repeat**
   $W_0 = \text{where}(W < m,\ 0.0,\ W);$
   $W_1 = \text{where}(W \leq l,\ \text{INF},\ W);$
   $W_2 = \text{where}(W > r,\ -\text{INF},\ W);$
   **if** $\text{sum}(W_0) \geq p$ **then**
      $l = m;$
   **else**
      $r = m;$
   **end if**
**until** $\text{max}(W_2) - \text{min}(W_1) \geq \epsilon$
Select indices $\mathcal{I}$ and set mask $\mathcal{M}$ where $W \geq l;$
**return** $\mathcal{I}, \mathcal{M};$

---

**Load Balancing with Awareness of Head Dynamism.** The top-$p$ Pruner enables head-wise dynamic budgets, but also raises load imbalance issues in the attention kernel. Traditional implementations allocate uniform computation resources to all heads. FlashInfer [52] deeply investigates this load imbalance problem, but only for requests with dynamic lengths. Twilight further reuses the load balancing algorithm in FlashInfer to address head-wise dynamism, by flattening the head dimension.

### 4.3 Overhead Analysis and Discussion

**Execution Time.** The execution time of Twilight consists of three parts according to the pipeline in Figure 5: $T_{\text{TokenSel}} + T_{\text{Pruner}} + T_{\text{SparseAttn}}$. Compared to the baseline sparse attention without Twilight, our method introduces an extra latency term $T_{\text{Pruner}}$ but reduces $T_{\text{SparseAttn}}$. Our hierarchical architecture naturally matches the hierarchical sparsity, where the number of tokens gradually decreases as the precision increases. Suppose the base algorithm in the Token Selector estimates token importance with a $1/16$ sparsity and/or precision reduction. Then the theoretical speedup can be formulated as $\frac{N/16+B_0}{N/16+B_0/4+B_1}$, where $B_0 = |\mathcal{I}_0|$ is the budget of the base Token Selector, and $B_1 = |\mathcal{I}_1|$ is the budget after pruned by Twilight with INT4. Assuming $B_0 = N/4$ and $B_1 = N/64$, the speedup would be approximately $2\times$. Here we omit the overheads of the top-$p$ kernel since SpGEMV dominates the latency when $B_0$ is around $N/8$ to $N/4$.

**Memory Overheads.** Twilight introduces an extra INT4 quantized K cache, which brings a $1/2 \times 1/4 = 1/8$ extra KV cache memory overhead. However, this additional cost does not appear in all cases. First, some base algorithms, like DS [50], already maintain an INT4 K cache. Second, some recent efforts have explored INT4 full attention [56]. This allows us to directly reuse the estimated attention weights calculated by the INT4 K cache in the attention computation, without maintaining the original FP16 K cache. Moreover, offloading and selective quantization (e.g., keeping the extra INT4 K cache only for hot tokens) can be leveraged if the GPU memory becomes a bottleneck, which we leave as future work.

**Integration with LLM Serving Systems.** Our system design naturally aligns with PagedAttention [20], so Twilight can be seamlessly integrated into popular serving systems like vLLM [20] and SGLang [60]. Other common techniques, such as prefix sharing and multi-phase attention [24, 60, 63, 51, 53], are also compatible with Twilight since we use page-level or token-level sparse operations, and can achieve a flexible computation flow.

## 5 Evaluation

In this section, we perform quantitative experiments to demonstrate that equipping state-of-the-art (SOTA) sparse attention algorithms with Twilight could improve efficiency while preserving accuracy. We present the accuracy and efficiency results in Section 5.1 and Section 5.2, respectively. At last, we perform ablation studies in Section 5.3.

## 5.1 Accuracy Evaluation

**Benchmarks and Models.** We evaluate Twilight on two types of benchmarks: long-context, which includes Longbench [1] and RULER [14], and medium-context (500 to 2k tokens), which includes GSM8K [4], COQA [33], and the perplexity on the PG-19 dataset [32]. We select three widely used models, Longchat-7B-v1.5-32k [22], LLaMA2-7B-Chat [40], and LLaMA-3.1-8B-Instruct [28] (128k context length), with two of them having long context ability $\geq$ 32k. They cover two mainstream attention implementations of multi-head attention (MHA) and group query attention (GQA) [28].

**Baselines.** We use two SOTA top-$k$ sparse attention methods, Quest [36] and DS [50], and one SOTA non-top-$k$ method, MagicPIG [3], as our baselines. Following the baselines, we do not apply any sparse methods to the first two layers to ensure fair comparison. For DS, we use the optimized configurations tuned for each model provided by its official repository. The hyperparameter $p$ of Twilight is set to 0.95 for LLaMA-2/3 and 0.85 for Longchat, which will be explored in Section 5.3. Note that MagicPIG does not employ the budget mechanism but instead relies on two configurable parameters, $K$ and $L$, which directly influence its accuracy. In our experiments, we adopt two standard configurations from the original MagicPIG paper. Due to the lack of official MagicPIG support for LLaMA-2, we exclude these experiments from our evaluation.

**Results on Longbench.** We comprehensively evaluate Twilight's long context ability on 12 different tasks chosen from Longbench, covering all task types, using two long-context models. For each top-$k$ baseline, we vary the budget from 256 to 8192, and then apply Twilight to dynamically determine the budget. We also equip "Full" with Twilight, in which the Token Selector is a trivial one that keeps all tokens.

The results are shown in Table 2. In Longchat, the Twilight framework is able to outperform its original version by up to 5.7% in the score, while successfully pruning up to **98%** of the redundant tokens overselected by the base algorithm. In LLaMA-3.1-8B-Instruct, Twilight achieves nearly zero accuracy loss ($<1\%$) with a slight increase in budget usage. We hypothesize that this slight increase is due to the knowledge being more compressed in LLaMA-3.1.

Table 2: Average scores on 12 different tasks from Longbench. We report relative error changes (improvement or degradation) when integrating Twilight with each base algorithm. Detailed results are in Table 5 in Appendix C.

| | Budget | Longchat-7B -v1.5-32k | LLaMA-3.1-8B -Instruct |
|---|---|---|---|
| Full | 32k | 36.78 | 52.01 |
| | **Twilight** | 38.52 (+4.7%) | 51.64 (-0.7%) |
| MagicPIG | K=8, L=75 | - | 51.70 |
| | K=10, L=150 | - | 51.32 |
| Quest | 256 | 31.26 | 38.20 |
| | 1024 | 36.85 | 47.79 |
| | 4096 | 37.33 | 50.79 |
| | 8192 | 37.10 | 51.44 |
| | **Twilight** | 38.04 (+2.5%) | 51.57 (+0.3%) |
| DS | 256 | 35.32 | 45.74 |
| | 1024 | 35.96 | 49.43 |
| | 4096 | 36.31 | 50.98 |
| | 8192 | 36.62 | 51.14 |
| | **Twilight** | **38.71** (+5.7%) | **51.73** (+1.2%) |

**Results on RULER.** We further evaluate Twilight on the RULER benchmark using the LLaMA-3.1-8B-Instruct model, which incorporates specialized tests including CWE/FWE for comprehensive non-retrieval accuracy evaluation. As presented in Table 3, while the standard Quest implementation underperforms the non-top-$k$ approaches, DS demonstrates surprisingly competitive results. When enhanced with Twilight, both variants show significant improvements: Quest-Twi achieves performance comparable to the SOTA non-top-$k$ method MagicPIG, while DS-Twi establishes new record-breaking performance, surpassing all existing methods.

**Results on Medium-Context Tasks.** We then demonstrate that the Twilight Pruner itself does not negatively impact performance on two zero-shot generation tasks, GSM8K and COQA using the lm-harness framework [10], as well as one perplexity test on the PG-19 dataset. Since we are specifically evaluating the Pruner, we do not integrate Twilight into the baseline models. All the baselines use a budget of 128, which is comparable to the budget after Twilight's pruning. The results in Table 4 show that Twilight outperforms Quest and DS by significant margins, with nearly zero loss compared to full attention.

## 5.2 Efficiency Evaluation

**Datasets.** We evaluate the efficiency of Twilight on both the self-attention operator and the end-to-end decoding stage on a single A100 GPU. We use Longbench, from which we select three different types

Table 3: Average scores on RULER.

|  | Budget | 16k | 32k | 64k | 96k | Avg. |
|---|---|---|---|---|---|---|
| Full | 100% | 92.88 | 89.42 | 85.17 | 85.23 | 88.18 |
|  | **Twilight** | 93.13 | 89.10 | 84.64 | 83.10 | 87.49 |
| MagicPIG | K=8, L=75 | 92.22 | **89.37** | 84.07 | 82.58 | 87.06 |
|  | K=10, L=150 | 91.38 | 88.20 | 83.34 | 82.02 | 86.23 |
| Quest | 4% | 79.35 | 79.8 | 78.64 | 73.22 | 77.75 |
|  | 8% | 87.31 | 83.06 | 80.82 | 75.28 | 81.62 |
|  | **Twilight** | 91.53 | 87.97 | 84.12 | **82.96** | 86.65 |
| DS | 4% | 92.04 | 88.11 | 84.43 | 82.56 | 86.79 |
|  | 8% | 92.89 | 88.70 | 84.39 | 82.72 | 87.18 |
|  | **Twilight** | **93.54** | 89.24 | **85.91** | 82.81 | **87.88** |

Table 4: Results on 3 medium-context benchmarks.

|  | GSM8K(flexible/strict)↑ | COQA(em/f1)↑ | PG-19 Perplexity↓ |
|---|---|---|---|
|  | LLaMA-2-7B-Chat | | |
| Full | 0.2290/0.2282 | 0.5935/0.7511 | 7.503 |
| Quest | 0.0523/0.0508 | 0.5710/0.7425 | 14.15 |
| DS | 0.2191/0.2190 | 0.5855/0.7401 | 7.622 |
| **Twilight** | **0.2153/0.2115** | **0.6088/0.7642** | **7.600** |
| *(Twilight Avg. Budget)* | 90.82 | 91.86 | 102.58 |
|  | LLaMA-3.1-8B-Instruct | | |
| Full | 0.7726/0.7475 | 0.6363/0.7882 | 7.490 |
| Quest | 0.3639/0.3533 | 0.6007/0.7554 | 19.00 |
| DS | 0.6194/0.6027 | **0.6455/0.7964** | 7.967 |
| **Twilight** | **0.7771/0.7604** | 0.6325/0.7869 | **7.529** |
| *(Twilight Avg. Budget)* | 112.40 | 86.85 | 110.98 |

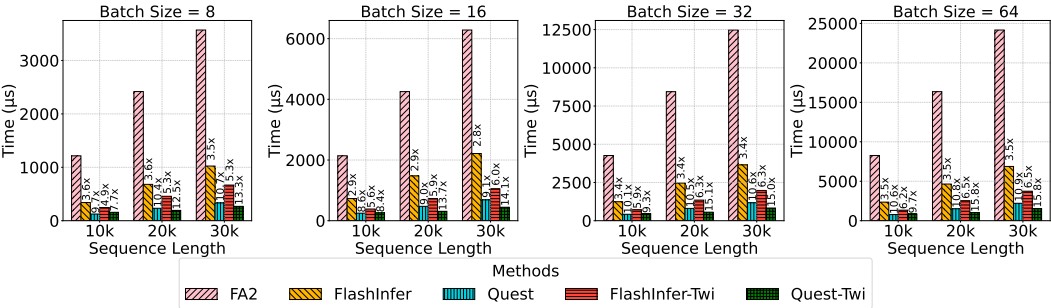

Figure 7: Latencies and speedups of self-attention at different sequence lengths and batch sizes.

of tasks: Qasper [6] for QA, GovReport [15] for summarization, and LCC [11] for coding. We use prompts ranging from 10k to 30k tokens for evaluation. Given that Twilight is designed for deploying sparse attention in LLM serving systems, we use batch inference in our experiments.

**Baselines and Implementation Details.** We compare our methods with the following baselines: PyTorch's scaled-dot-product-attention (SDPA), with **FlashAttention2** (FA2) [5] and Memory-Efficient Attention [21] as the backends; **FlashInfer** [52], a high-performance kernel library for LLM serving; **Quest**, which achieves SOTA runtime performance among sparse attention methods. We integrate Twilight with both FlashInfer and Quest, resulting in **FlashInfer-Twi** and **Quest-Twi**. We modify the Quest kernels to support batch inference. We implement Twilight using both CUDA and OpenAI Triton [39], following the technical details described in Section 4.2.

**Self-Attention Speedup.** We first evaluate the speedups on the self-attention operator across different batch sizes and sequence lengths. As Figure 7 shows, FlashInfer-Twi and Quest-Twi achieve speedups up to $6.5\times$ and $15.8\times$ compared with FlashAttention2. Moreover, they accelerate the respective base algorithms FlashInfer and Quest by $2.4\times$ and $1.4\times$.

**End-to-End Decoding Speedup.** We evaluate end-to-end decoding with batch sizes ranging from 32 to 256 for various serving scenarios. Figure 8 illustrates that Quest-Twi achieves up to a $3.9\times$ speedup compared with FlashInfer, and a $1.35\times$ speedup compared to Quest without Twilight.

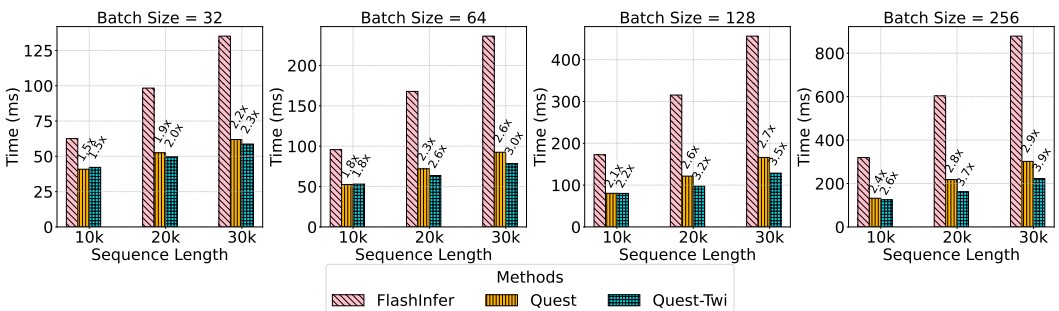

Figure 8: Time-Per-Output-Token (TPOT) improvements in end-to-end serving scenarios.

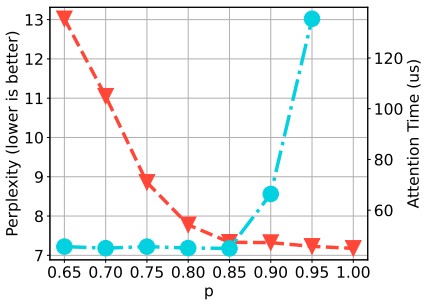

Figure 9: PG-19 perplexity (accuracy) and sparse attention latency (efficiency) under different threshold $p$ values.

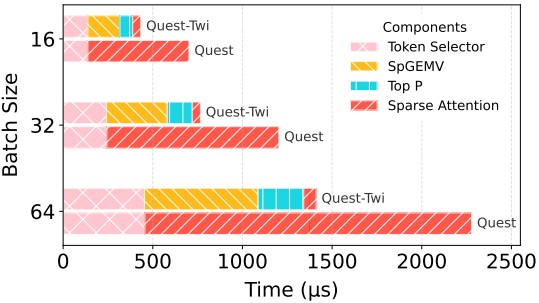

Figure 10: Time breakdown of self-attention. At batch size 64, Quest-Twi outperforms Quest by about $2\times$.

## 5.3 Ablation Study

**Sensitivity to Threshold $p$.** Notably, although we introduce the threshold $p$ in order to get rid of the budget $k$, we argue that $p$ is a more reasonable and tunable hyperparameter. This is because $p$ represents the accumulated probability, which is less influenced by the different distributions that may occur for different heads/layers/queries. In contrast, $k$ is highly sensitive to different distributions, as illustrated in Figure 1. This allows us to simply tune $p$ for a fixed model, in a way such as calibrating with a small dataset.

For the impact of $p$ on model accuracy, we test the perplexity on the PG-19 dataset when using different thresholds $p$. For the impact on runtime efficiency, the $p$ value directly controls the pruning aggressiveness and affects the attention time via the pruned token number. We evaluate the sparse attention kernel speed after pruned on the TrivialQA dataset. As Figure 9 shows, the accuracy and efficiency strike a balance at $p \approx 0.85$, making us choose $p = 0.85$ for Longchat-7B-v1.5-32k.

**Time Breakdown for Twilight.** Given Twilight's hierarchical architecture, which comprises three distinct components, it is insightful to analyze the execution time breakdown to further understand the benefit and cost. Figure 10 illustrates the time breakdown for different batch sizes in a 32k retrieval task. In this scenario, Quest employs a budget of 8192 (1/4 sparsity), while Twilight further prunes this budget down to 256. The breakdown aligns closely with the theoretical cost model presented in Section 4.3, demonstrating that Twilight significantly reduces the time required for the sparse attention kernel while introducing minor overheads.

## 6 Conclusion

In this paper, we first highlight that existing top-$k$ sparse attention methods struggle to find optimal budgets due to the dynamic nature of attention weight distributions. We then introduce Twilight, a framework with a hierarchical select-then-prune architecture that leverages top-$p$ sampling to address this issue. Twilight can adaptively prune up to 98% tokens, resulting in a $15.4\times$ speedup for the self-attention operator and a $3.9\times$ reduction in the end-to-end per-token latency. Comparing to the base sparse attention algorithm it is applied to, Twilight offers an additional $1.4\times$ speedup. Our work underscores the importance of adaptive attention sparsity, and paves a promising way for future research on sparse attention mechanisms.

## Acknowledgment

The authors thank the anonymous reviewers for their valuable suggestions, Yilong Zhao for helping us on kernel optimization, and the Tsinghua IDEAL group members for constructive discussion. Mingyu Gao is the corresponding author.

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

# A  Budget Dynamism at Different Levels

As introduced in Section 1, various levels of budget dynamism exist. We propose and analyze four distinct levels of KV cache budget dynamism, as illustrated in Figure 11. They are prompt-wise, query-wise, layer-wise, and head-wise dynamism.

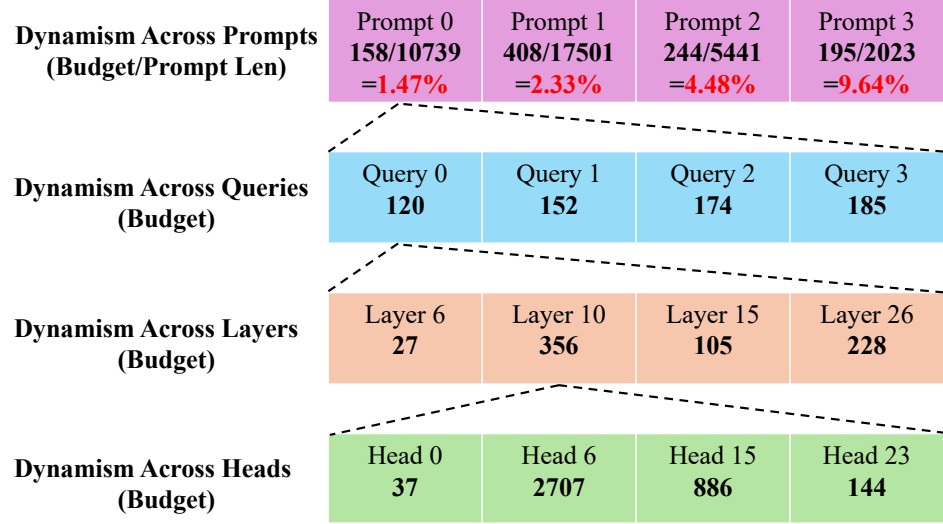

Figure 11: Budget dynamism observed in oracle top-$p$ attention. We observe the dynamism across four dimensions: different **prompts (tasks)**, different **queries** within the same prompt, different **layers** in the same query, and different **heads** in the same layer.

Table 5: **Full results on Longbench.** The highest score in each task (except for "Full") is marked in bold. The average budget after Twilight's pruning is shown following the method name. We also report the relative error changes (improvement or degradation) when integrating Twilight with each base algorithm.

| Method | Budget | Single-Doc. QA | | Multi-Doc. QA | | | Summarization | | | Few-shot | Synthetic | Code | | Avg. Score |
|---|---|---|---|---|---|---|---|---|---|---|---|---|---|---|
| | | Qasper | MF-en | HotpotQA | 2WikiMQA | Musique | GovReport | QMSum | MultiNews | TriviaQA | PR-en | LCC | Repobench-P | |
| | | | | | | | Longchat-7B-v1.5-32k | | | | | | | |
| Full | 32k | 29.48 | 42.11 | 30.97 | 23.74 | 13.11 | 31.03 | 22.77 | 26.09 | 83.25 | 30.50 | 52.70 | 55.62 | 36.78 |
| | Twilight (Avg. 146) | 31.74 | **43.91** | 33.59 | **25.65** | **13.93** | 32.19 | **23.15** | 26.30 | 85.14 | 34.50 | 54.98 | 57.12 | 38.52 (+4.7%) |
| Quest | 256 | 26.00 | 32.83 | 23.23 | 22.14 | 7.45 | 22.64 | 20.98 | 25.05 | 67.40 | 33.60 | 48.70 | 45.07 | 31.26 |
| | 1024 | 31.63 | 42.36 | 30.47 | 24.42 | 10.11 | 29.94 | 22.70 | 26.39 | 84.21 | 34.5 | 51.52 | 53.95 | 36.85 |
| | 4096 | 29.77 | 42.71 | 32.94 | 23.94 | 13.24 | 31.54 | 22.86 | 26.45 | 84.37 | 31.50 | 53.17 | 55.52 | 37.33 |
| | 8192 | 29.34 | 41.70 | 33.27 | 23.46 | 13.51 | 31.18 | 23.02 | 26.48 | 84.70 | 30.00 | 53.02 | 55.57 | 37.10 |
| | Twilight (Avg. 131) | 31.95 | 43.28 | 31.62 | 24.87 | 13.48 | **32.21** | 22.79 | 26.33 | 84.93 | 33.50 | 54.86 | 56.70 | 38.04 (+2.5%) |
| DS | 256 | 28.28 | 39.78 | 27.10 | 20.75 | 9.34 | 29.68 | 21.79 | 25.69 | 83.97 | 32.00 | 52.01 | 53.44 | 35.32 |
| | 1024 | 30.55 | 41.27 | 30.85 | 21.87 | 7.27 | 26.82 | 22.95 | 26.51 | 83.22 | 31.50 | 53.23 | 55.50 | 35.96 |
| | 4096 | 28.95 | 41.90 | 32.52 | 23.65 | 8.07 | 29.68 | 22.75 | **26.55** | 83.34 | 30.00 | 52.77 | 55.48 | 36.31 |
| | 8192 | 29.05 | 41.42 | 31.79 | 22.95 | 12.50 | 30.44 | 22.50 | 26.43 | 83.63 | 30.50 | 52.87 | 55.33 | 36.62 |
| | Twilight (Avg. 126) | **32.34** | 43.89 | **34.67** | 25.43 | 13.84 | 31.88 | 23.01 | 26.32 | **85.29** | 35.50 | 55.03 | **57.27** | **38.71** (+5.7%) |
| | | | | | | | LLaMA-3.1-8B-Instruct | | | | | | | |
| Full | 128k | 46.17 | 53.33 | 55.36 | 43.95 | 27.08 | 35.01 | 25.24 | 27.37 | 91.18 | 99.50 | 62.17 | 57.76 | 52.01 |
| | Twilight (Avg. 478) | 43.08 | 52.99 | 52.22 | 44.83 | 25.79 | 34.21 | **25.47** | 26.98 | 91.85 | 100.00 | 64.06 | 58.22 | 51.64 (-0.7%) |
| MagicPIG | K=8, L=75 | **45.03** | 54.24 | **56.46** | **47.34** | 26.58 | 33.63 | 24.98 | 26.70 | 92.13 | **100.00** | 61.94 | 51.40 | 51.70 |
| | K=10, L=150 | 44.68 | 53.63 | 56.19 | 47.18 | **26.79** | 33.31 | 25.13 | 26.22 | 91.89 | 99.50 | 60.07 | 51.15 | 51.32 |
| Quest | 256 | 24.65 | 37.50 | 30.12 | 23.60 | 12.93 | 27.53 | 20.11 | 26.59 | 65.34 | 99.50 | 49.70 | 45.27 | 38.20 |
| | 1024 | 38.47 | 49.32 | 47.43 | 38.48 | 20.59 | 33.71 | 23.67 | 26.60 | 81.94 | 99.50 | 60.78 | 52.96 | 47.79 |
| | 4096 | 43.97 | 53.64 | 51.94 | 42.54 | 24.00 | 34.34 | 24.36 | 26.75 | 90.96 | 99.50 | 62.03 | 55.49 | 50.79 |
| | 8192 | 44.34 | 53.25 | 54.72 | 44.84 | 25.98 | 34.62 | 24.98 | 26.70 | 91.61 | 100.00 | 62.02 | 54.20 | 51.44 |
| | Twilight (Avg. 427) | 43.44 | 53.2 | 53.77 | 43.56 | 25.42 | 34.39 | 25.23 | 26.99 | 91.25 | 100.0 | 63.55 | 58.06 | 51.57 (+0.3%) |
| DS | 256 | 38.24 | 49.58 | 43.38 | 31.98 | 15.52 | 33.40 | 24.06 | 26.86 | 84.41 | 99.50 | 53.28 | 48.64 | 45.74 |
| | 1024 | 42.97 | 54.65 | 51.75 | 33.92 | 20.39 | 34.50 | 24.92 | 26.71 | **92.81** | 99.50 | 62.66 | 48.37 | 49.43 |
| | 4096 | 43.50 | 53.17 | 54.21 | 44.70 | 23.14 | **34.73** | 25.40 | 26.71 | 92.78 | 99.50 | 62.59 | 51.31 | 50.98 |
| | 8192 | 43.82 | 53.71 | 54.19 | 45.13 | 23.72 | 34.27 | 24.98 | 26.69 | 91.61 | **100.00** | 62.40 | 52.87 | 51.14 |
| | Twilight (Avg. 446) | 43.08 | 52.89 | **54.68** | 44.86 | 24.88 | 34.09 | 25.20 | **27.00** | 91.20 | 100.00 | 63.95 | **58.93** | **51.73** (+1.2%) |

# B Kernel Implementation Details

## B.1 Implementation of Mixed-Precision SpGEMV

**Calculation Process.** As outlined in Section 4.2, our implementation requires a GEMV kernel that computes the product of an FP16 query vector and an INT4 quantized key matrix ($\mathbf{q}$fp16·$\mathbf{K}$int4) with paged indexing. We adapt the attention decoding kernel from FlashInfer [52] for this purpose. The kernel execution follows two main steps: (1) asynchronously loading and dequantizing the quantized K cache from the global memory into the shared memory using `cp.async`; and (2) computing the dot product. To mitigate long memory latency, we employ a two-stage pipeline that overlaps the data loading of a subsequent

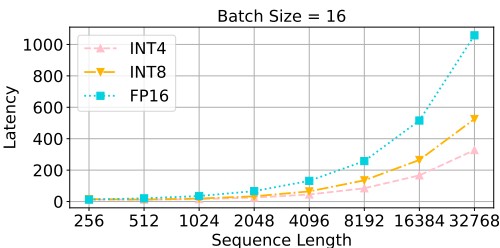

Figure 12: SpGEMV operator latency with different quantization bits.

block with the computation of the current block. We use FP16 to store the dequantized K cache instead of FP32 to optimize the computation given that such accuracy tradeoff is tolerable as a score estimator.

**Dequantization.** Following the design of QServe [25], we employ *per-head, dynamic* KV quantization and store the FP16 `scale` and `zero` for each head using the same paged memory layout as the K cache. The $\mathbf{K}$ matrix is dequantized on-the-fly using per-head quantization parameters (`scale` and `zero`). Our dequantization routine builds upon the fast algorithm from [19] (as implemented in NVIDIA's FasterTransformer [31]), which utilizes custom PTX assembly instructions for efficient type conversion between INT4 and FP16.

**Bit-packing.** INT4 $\mathbf{K}$ elements are packed into an `uint8_t` buffer, with two 4-bit elements stored within each 8-bit byte — this aligns with the byte-addressable nature of C++. To simplify the dequantization logic, we first add an offset of +128 to each INT4 element, converting it to an unsigned value, before packing them in an interleaved manner. Address calculation for this packed buffer is remapped to stride at a 4-bit granularity; this is achieved by halving the effective byte offset [59].

We conduct an ablation study on the impact of quantization bits on the efficiency as Figure 12 shows. Our optimizations on the dequantization and dot product make the operator memory-bound and thus benefit from quantization.

## B.2 Tackling Group Query Attention

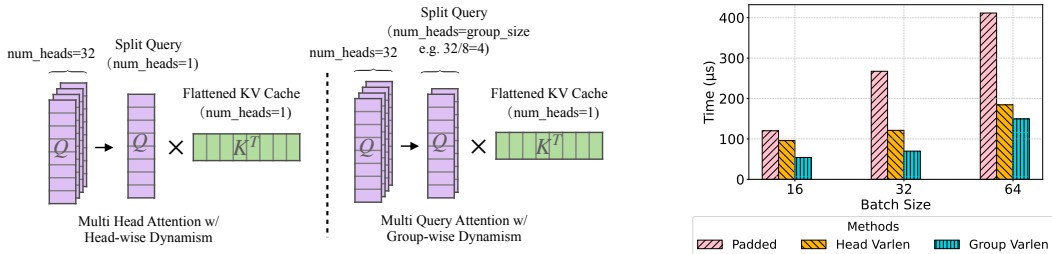

Figure 13: **Left:** Head-wise/group-wise varlen attention with flattend paged KV cache in Twilight. **Right:** Comparison among the three attention methods on a real budget distribution of a LLaMA-3.1-7B layer on a 16k retrieval task. Here "Padded" means padding all heads to the maximum budget length; "Head Varlen" loads KV at the head granularity which causes repeated loading; and "Group Varlen" strikes a balance between the two methods.

Group Query Attention (GQA) [28], a technique widely adopted in recent model architectures like LLaMA 3, maps a group of query heads to a single key-value head. This structure, however, is inherently incompatible with query-aware sparse attention. The incompatibility arises because query-aware sparse attention relies on individual query vectors to identify important tokens, but

GQA creates a mismatch at the granularity of attention heads. A brute-force solution would be to load tokens independently for each query head, but this leads to inefficient, repeated memory reads. Twilight addresses this issue by operating at the granularity of query groups. Specifically, the set of tokens selected for a given query group is the union of tokens identified by all query heads within that group [54].

As discussed in Section 4.2, our top-$p$ attention mechanism natively supports head-wise dynamism. However, when integrated with GQA, this head-wise dynamism inherently transitions to group-wise dynamism, meaning that all heads within the same group share a common token budget. Figure 13 shows our attention design with flattened paged KV cache, which supports head-wise varlen attention for MHA and group-wise varlen attention for GQA. This design represents a deliberate trade-off, balancing implementation efficiency with compatibility for modern attention algorithms. We also compare the efficiency of the three different attention implementations in Figure 13.

## C  Full Results on Longbench

Please refer to Table 5.

## D  Accuracy Comparison with Token Dropping Methods

As discussed in Section 2, top-$k$ sparse attention methods can be broadly categorized into two types: token dropping and token selecting. Prior research [36] has established that token selecting generally outperforms token dropping, as the latter inevitably incurs irreversible information loss. To further validate this observation, we conduct comparative experiments between Twilight and two representative token-dropping methods: StreamingLLM [46] and SnapKV [23]. As demonstrated in Table 6, DS-Twilight achieves notably better performance over both baseline methods.

Table 6: Comparison of StreamingLLM, SnapKV, and Twilight on the Longbench benchmark with the Longchat-7B-v1.5-32k model.

| Dataset | StreamingLLM (Budget=4096) | SnapKV (Budget=4096) | DS-Twilight |
|---------|---------|---------|---------|
| Qasper | 26.39 | 29.44 | **32.34** |
| MulQA-en | 33.2 | 40.03 | **43.89** |
| HotpotQA | 24.29 | 33.67 | **34.67** |
| 2WikiMQA | 20.1 | 24.13 | **25.43** |
| Musique | 10.87 | 13.45 | **13.84** |
| GovReport | 26.92 | 26.09 | **31.88** |
| QMSum | 20.8 | 22.53 | **23.01** |
| MultiNews | 26.46 | 25.61 | **26.32** |
| TrivialQA | 75.6 | 80.82 | **85.29** |
| PR-en | 24.17 | 30.25 | **35.50** |
| LCC | 52.47 | 52.62 | **55.03** |
| Repobench-P | 51.02 | 55.99 | **57.27** |
| Avg. | 32.69 | 36.22 | **38.71** |

## E  Efficiency Evaluation in Offloading Scenarios

Notably, in memory-offloading scenarios where the per-token loading cost dominates, Twilight could achieve more significant gains. This is because Twilight reduces the number of loaded tokens with a fixed estimation cost. Table 7 shows Twilight could achieve up to $16\times$ speedups compared to Quest.

Table 7: Latency (in microseconds) of a single attention operator in offloading scenarios, where corresponding tokens in the KV cache are loaded from the CPU memory.

| | 10k | 20k | 30k |
|---|---|---|---|
| Quest | 3038.98 | 5990.75 | 8490.95 |
| Quest-Twi | 415.89 | 480.61 | 527.77 |

# F   Limitations and Future Work

While Twilight effectively accelerates existing top-$k$ sparse attention methods, our analysis in Figure 10 reveals non-negligible estimation overheads. This makes Twilight particularly advantageous in scenarios like serving with large batch sizes or offloading, where the cost of loading tokens from the KV cache dominates. Section B.2 shows head-wise dynamism is unfriendly with GQA, which leads to some challenges to integrate Twilight with new model architectures. Future research could focus on optimizing the estimation method to further improve the end-to-end latency and throughput, and how to integrate Twilight with other model architectures like multi-head latent attention (MLA) [7].

