# OpenReview forum: "Twilight: Adaptive Attention Sparsity with Hierarchical Top-$p$  Pruning"
_NeurIPS.cc/2025/Conference — NeurIPS 2025 spotlight_

### Official Review · Reviewer_Pft7 · 2025-07-01

**Clarity:** 3
**Significance:** 3
**Originality:** 3
**Rating:** 5
**Confidence:** 4

**Summary:**

Twilight proposes an adaptive framework for long-context large language models (LLMs) to address the limitations of static budget settings in traditional sparse attention methods. By leveraging top-p sampling (nucleus sampling), Twilight dynamically prunes tokens to maintain a cumulative attention weight threshold (e.g., 90%), overcoming over/under-selection issues in fixed top-k approaches. The hierarchical "select-then-prune" architecture first uses base algorithms (e.g., Quest/DS) to select tokens conservatively, then refines via top-p with 4-bit quantization and GPU kernel optimizations. Experiments show Twilight achieves up to 1.4× speedup over state-of-the-art sparse attention, prunes 98% redundant tokens, and maintains near-zero accuracy loss across mid/long-context benchmarks (Longbench, RULER)

**Questions:**

Mention in Weakness

**Ethical Concerns:**

["NO or VERY MINOR ethics concerns only"]

**Final Justification:**

I acknowledge the limitation and look forward to a future fix. I will maintain my score.

**Limitations:**

yes

**Quality:**

3

**Strengths And Weaknesses:**

Strengths
1. Adaptive Budgeting with Top-p Sampling Replaces static top-k with dynamic top-p sampling, which adjusts the token budget based on the cumulative attention weight sum.
2. Significant Efficiency Gains. Achieves up to 1.4× speedup over state-of-the-art sparse attention methods (e.g., Quest, DS) for self-attention, and end-to-end decoding speedup. Reduces memory access and computation by dynamically pruning redundant tokens, especially in long-context scenarios.
3. System-Level Optimizations. Implements efficient kernels (e.g., SpGEMV with 4-bit quantization) and load balancing for head-wise dynamism, reducing latency and memory overhead.

Weakness:
1. The top-p pruning introduces an additional computation step, which may offset speed gains in scenarios with small batch sizes or minimal token pruning.

---

> ### Author Rebuttal · Authors · 2025-07-31
>
> Thanks for your detailed review and insightful advice. Below, we address each concern and outline planned revisions to strengthen the paper.
>
> **1. Additional Computation Cost of Top-p Pruning**
>
> We sincerely appreciate the reviewer's insightful observation regarding the Pruner's computational overhead. As noted in Appendix E, we acknowledge that the current estimation process introduces non-trivial latency that can diminish speed benefits in certain edge cases, like the reviewer mentions, small batch sizes or scenarios with minimal token pruning. While we acknowledge the current computational overhead in the Pruner as noted by the reviewer, we maintain that Twilight's core contributions remain significant for the top-p sparse attention and hierarchical selector-pruner framework.

---

> > ### Comment · Reviewer_Pft7 · 2025-08-05
> >
> > Thanks for clarifying. I acknowledge the limitation and look forward to a future fix. I will maintain my score.

---

> > > ### Author Response · Authors · 2025-08-06
> > >
> > > Thanks for acknowledging my clarification regarding the limitations. Have a nice day!

---

### Official Review · Reviewer_QNJM · 2025-07-02

**Clarity:** 2
**Significance:** 2
**Originality:** 2
**Rating:** 5
**Confidence:** 4

**Summary:**

The paper introduces Twilight, a select-then-prune framework that upgrades any existing top-k sparse-attention scheme with a top-p pruning stage. A base token-selector first picks a conservative subset of KV positions; a lightweight pruner then keeps the smallest subset whose estimated attention weights sum to a user-set threshold p. Across LongBench, RULER, GSM8K, COQA and PG-19, Twilight reportedly prunes up to 98 % of tokens with almost no accuracy loss and delivers 15.4x operator-level and 3.9x E2E speed-ups, or ~ 1.4 times beyond the best sparse baselines.

**Questions:**

1. The INT4 cache adds ~12 % KV footprint. Is it possible to quantify the wall-clock impact compared to a top-k based method which does not incur additional memory footprint?
2. How sensitive is the method to different `p` thresholds? The paper mentions usage of different values for different models.
3. Binary search still needs repeated sum() reductions. Could reservoir sampling or approximate CDF estimates cut this further?

**Ethical Concerns:**

["NO or VERY MINOR ethics concerns only"]

**Final Justification:**

The authors addressed my concerns about sensitivity to parameter `p` with an ablation study, which makes the submission more sound. Unaddressed concerns is that limitations (KV cache overhead) could be stated more clearly in the main part of the paper, instead of appendix. As the rebuttal addresses some of my technical doubts about the approach, I will raise the score.

**Limitations:**

No limitations section - would be good to include some of the ones mentioned in "Weaknesses" section - boundedness by the base token selector and KV overhead.

**Paper Formatting Concerns:**

no major concerns

**Quality:**

3

**Strengths And Weaknesses:**

Strengths:
* The idea and motivation of using top-p instead of commonly used top-k is natural and intuitive.
* Significant latency gains compared to the baselines.

Weaknesses:
* The method efficacy is bounded by the base token-selector.
* Extra INT4 copy increases KV memory by up to 12.5 % and is always resident on-device

---

> ### Author Rebuttal · Authors · 2025-07-31
>
> Thanks for your detailed review and insightful advice. Below, we address each concern and outline planned revisions to strengthen the paper.
>
> **1. Sensitivity to Hyperparameter p**
>
> We appreciate the reviewer's valuable suggestion regarding the efficiency analysis of parameter p. In our current Figure 6, we have shown the impact of p on model performance (accuracy). The following table shows the impact on efficiency evaluated on the TrivialQA dataset. We will augment this analysis in the camera-ready version.
>
> | p | 0.65  | 0.70  | 0.75  | 0.80  | 0.85  | 0.90  | 0.95  |
> |------|-------|-------|-------|-------|-------|-------|-------|
> | PG-19 Perplexity (Lower is better)  | 13.021| 11.050| 8.857 | 7.777 | 7.332 | 7.324 | 7.228 |
> | Pruned Tokens Number  | 72 | 93 | 117 | 160 | 237 | 381 | 807 |
> | Sparse Attn Time bs=32 (us)  | 45.63 | 45.01 | 45.62 | 45.06 | 44.93 | 66.38 | 135.46 |
> | Sparse Attn Time bs=64 (us)  | 46.60 | 45.90 | 54.93 | 64.88 | 84.32 | 124.68 | 243.84 |
>
> In summary, the `p` value directly controls pruning aggressiveness and affects the time of the Sparse Attention stage by the pruned tokens number. We will expand this analysis in the ablation study part of the camera-ready version.
>
> Notably, although we introduce the hyperparameter `p` in order to get rid of `k`, `p` is a more reasonable and tunable hyperparameter. This is because `p` represents the accumulated probability, which is less impacted by the different distributions that may occur for different heads/layers/queries. In contrast, `k` is highly sensitive to different distributions, as illustrated in Figure 1 in the paper. This allows us to just tune `p` for a fixed model. We can use a small dataset to calibrate it.
>
>
> **2. Overhead of INT4 KV Cache**
>
> We fully acknowledge the overhead introduced by INT4 KV Cache in Twilight. As the reviewer correctly identified, this manifests in two key dimensions: (1) Storage overhead, as the reviewer mentioned, 12.5% extra overhead to GPU memory; (2) I/O overhead, which occurs during the attention forward process.
>
> The wall-clock impact, as far as we understand, refers to the I/O overhead where the SpGEMV needs to read corresponding INT4 KV blocks to estimate the attention score. This can be found in the breakdown analysis of Figure 7, where “SpGEMV” part reflects this overhead. Note that SpGEMV is a memory-bound kernel like the Attention computation. Hence the total kernel time is roughly equal to the read time.
>
> To deal with the Storage Overhead (on-device INT4 KV Cache), offloading may be a possible solution. But we find it is more suitable to offload the full precision KV instead of INT4 KV as the pruned tokens number (B1, follows notations in Section 4.3) is far less than the initial budget (B0). We discuss this potential scenario in Appendix D.
>
> **3. Improve Binary Search during Sampling**
>
> We sincerely appreciate the reviewer's valuable suggestion regarding alternative sampling approaches. After careful experimentation on the approach of CDF estimation, we find it is unsuitable for Twilight's architecture because our selector-pruner architecture requires higher precision. A concurrent work that also highlights the potential of top-p attention, Tactic (cited in our paper), makes this method work upon full attention.
>
> For random methods like reservoir sampling, it is worth mentioning that MagicPIG, a work which is evaluated and compared with Twilight, uses another method called LSH sampling. Twilight tends to adopt non-randomized algorithms to maintain stability as an optimizer.
>
> However, we fully acknowledge the optimization potential in our current binary search implementation and are actively exploring improvements. Notably, the FlashInfer community's recent breakthrough in sorting-free GPU sampling kernels presents a promising solution, as documented in their technical blog. Their novel Dual Pivot Rejection Sampling algorithm achieves an approximately 2x speedup over traditional binary search. We are trying to integrate it with Twilight and will incorporate this collaborative advancement in our camera-ready version.
>
> **4. Section of Limitations**
>
> We discuss the limitations in Appendix E, as required by the submission guidelines.

---

### Official Review · Reviewer_HTDe · 2025-07-04

**Clarity:** 3
**Significance:** 4
**Originality:** 3
**Rating:** 4
**Confidence:** 4

**Summary:**

This work proposes one idea, that is to replace the top-k strategy used in current sparse attention methods with top-p. The idea is based on the observation that different attention heads can produce very different probability distributions, with some attending to a few tokens and others having more flat attending behaviors. In this case, static top-k strategy can either over-select or under-select in varying scenarios. In contrast, top-p selection has a more normalized behavior that is self-adaptive to different distributions on attention heads. This is similar to the top-p sampling in LLMs. While the idea is simple and directly applicable to existing sparse attention methods, there are certain architecture designs that need improved for efficient implementation, which is the other major contributions of this work. These include efficient sparse GEMV with 4-bit quantization of key cache and efficient, parallel top-p selection with binary search.

The empirical experiments consist of three parts, effectiveness, efficiency and ablation study. The first part shows improved accuracies over the base methods on different types of tasks. The second part breaks the efficiency benefits of the proposed method. The ablation study investigates the influence of the threshold p and the cost breakdown.

**Questions:**

- I wonder why the time spent for sparse attention in Quest-Twi is so much shorter than that in Quest? Is that because of the average number of tokens selected by Quest-Twi is much lower than that by Quest?

**Ethical Concerns:**

["NO or VERY MINOR ethics concerns only"]

**Final Justification:**

The authors' responses addressed many of my concerns. I remain positive towards this work.

**Limitations:**

yes

**Quality:**

3

**Strengths And Weaknesses:**

Strengths:
+ Simple yet effective idea, which successfully borrows the idea of top-p sampling.
+ The authors successfully identify the difficulties of apply top-p sampling to existing sparse attention methods and address them well.
+ Impressive performance with further reduced computation cost.

Weaknesses:
- Since replacing top-k with top-p is a straightforward idea, the technical contributions of this work mostly lie at the how the implementation issues are solved with most efficiency. However, the ablation study fails to do so. We are not clear how important the "efficient" kernel implementation described in Section 4.2 is for the efficiency shown in the experiments.
- It is not clear how sensitive the efficiency is to the value of p. Figure 6 only shows the influence on the model performance. Also, Figure 6 has the wrong label on x-axis.

---

> ### Author Rebuttal · Authors · 2025-07-31
>
> Thanks for your detailed review and insightful advice. Below, we address each concern and outline planned revisions to strengthen the paper.
>
> **1. Detailed Description of Efficient Kernel implementation**
>
> We fully agree with the reviewer's assessment regarding the importance of our kernel implementations to the overall contribution. While space constraints limited our presentation of these technical details in the current version, we will significantly expand the kernel implementation documentation in the camera-ready version, including warp-level reductions, better utilization of the memory bandwidth, fusion of dequantization and SpGEMV, details on head-wise load balancing, etc.
>
> **2. The Impact of p on Efficiency**
>
> We sincerely thank the reviewer for catching the incorrect x-axis label in Figure 6, which has now been corrected. Regarding the impact of the `p` value on computational efficiency, the following shows a basic statistics evaluated on the TrivialQA dataset:
>
> | p | 0.65  | 0.70  | 0.75  | 0.80  | 0.85  | 0.90  | 0.95  |
> |------|-------|-------|-------|-------|-------|-------|-------|
> | PG-19 Perplexity (Lower is better)  | 13.021| 11.050| 8.857 | 7.777 | 7.332 | 7.324 | 7.228 |
> | Pruned Tokens Number  | 72 | 93 | 117 | 160 | 237 | 381 | 807 |
> | Sparse Attn Time bs=32 (us)  | 45.63 | 45.01 | 45.62 | 45.06 | 44.93 | 66.38 | 135.46 |
> | Sparse Attn Time bs=64 (us)  | 46.60 | 45.90 | 54.93 | 64.88 | 84.32 | 124.68 | 243.84 |
>
>
> In summary, the `p` value directly controls pruning aggressiveness and affects the time of the Sparse Attention stage by the pruned tokens number. We will expand this analysis in the ablation study part of the camera-ready version.
>
>
> Notably, although we introduce the hyperparameter `p` in order to get rid of `k`, `p` is a more reasonable and tunable hyperparameter. This is because `p` represents the accumulated probability, which is less impacted by the different distributions that may occur for different heads/layers/queries. In contrast, `k` is highly sensitive to different distributions, as illustrated in Figure 1 in the paper. This allows us to just tune `p` for a fixed model. We can use a small dataset to calibrate it.
>
>
> **3. Question on Time Breakdown**
>
> Thanks for your interesting question! Your understanding is fully correct. And as Table 5 shows, Twilight can prune the original budget (~8192) to extremely low (100 to 400), which largely reduces the time of the Sparse Attention part.

---

### Official Review · Reviewer_mTkt · 2025-07-06

**Clarity:** 3
**Significance:** 2
**Originality:** 3
**Rating:** 4
**Confidence:** 4

**Summary:**

The paper proposes Twilight, a framework that enhances existing sparse attention algorithms by replacing top-k token selection with top-p sampling to dynamically adjust the KV cache budget. The authors argue that this approach addresses the limitations of fixed budgets in sparse attention mechanisms, leading to improved efficiency and minimal accuracy loss. The framework is evaluated on both short- and long-context benchmarks, demonstrating speedups over existing methods while maintaining competitive accuracy. However, the technical contribution is incremental, and several methodological and experimental aspects raise concerns.

**Questions:**

See my above comments.

**Ethical Concerns:**

["NO or VERY MINOR ethics concerns only"]

**Final Justification:**

The authors address all of my concerns.

**Limitations:**

See my above comments.

**Quality:**

2

**Strengths And Weaknesses:**

**Positive points**

1.	The paper clearly identifies the limitations of static budget allocation in sparse attention mechanisms and provides a well-motivated rationale for adopting top-p sampling.
2.	The authors conduct extensive experiments across multiple benchmarks and models, demonstrating the generalizability of their approach.
3.	The paper is well-structured, with clear figures and tables that aid understanding. The hierarchical pruning architecture is described in sufficient detail.

**Negative points**

1.	The paper’s core contribution is incremental—replacing top-k selection with top-p in existing sparse attention methods. This adaptation lacks significant novelty, as it does not introduce a fundamentally new mechanism.
2.	When integrated with Quest/DS, Twilight prunes more tokens yet improves performance. If this is due to better removal of irrelevant tokens, why does Twilight not outperform full attention (which retains all tokens)? The authors must clarify this contradiction, as it suggests unresolved trade-offs in the pruning strategy.
3.	The performance improvements on long-context tasks are limited. While Twilight shows improvements over Quest/DS, its performance on long-context benchmarks (e.g., Longbench, RULER) remains inferior to MagicPIG.
4.	In short task evaluations, the authors state that baselines use a fixed budget of 64, which is claimed to be comparable to Twilight’s pruned budget. However, the average post-pruning budget for Twilight is not reported, making it impossible to verify this claim.
5.	Recent advances in sparse attention (e.g., [A-E]) are neither compared nor discussed. Position the method against current state-of-the-art techniques.

**Reference**

[A] MInference 1.0: Accelerating Pre-filling for Long-Context LLMs via Dynamic Sparse Attention, NeurIPS 2024.

[B] Core Context Aware Transformers for Long Context Language Modeling. ICML 2025

[C] Flexprefill: A Context-Aware Sparse Attention Mechanism for Efficient Long-Sequence Inference. ICLR 2025

[D] MMInference: Accelerating Pre-filling for Long-Context VLMs via Modality-Aware Permutation Sparse Attention. ICML 2025

[E] Curse of High Dimensionality Issue in Transformer for Long Context Modeling. ICML 2025.

---

> ### Author Rebuttal · Authors · 2025-07-31
>
> Thanks for your detailed review and insightful advice. Below, we address each concern and outline planned revisions to strengthen the paper.
>
> **1. Comparing with Recent Sparse Attention Works**
>
> We sincerely appreciate the reviewer's valuable suggestions regarding prior work comparisons. We have already compared Twilight with two SOTA **top-k decoding methods** Quest/DoubleSparse and one SOTA **non-top-k decoding method** MagicPIG. But we acknowledge that our current presentation may have caused some confusion about our methodological focus. To clarify: Twilight specifically optimizes **the decoding stage**, which dominates the computations in emerging workloads like reasoning and conversation, while works [A-E] primarily target **the prefill stage**, addressing different optimization challenges. Our approach is fundamentally orthogonal to such prefill-stage optimizations. We will enhance the Related Work section to clearly delineate these two optimization domains and better position Twilight within the decode-stage optimization landscape.
>
> **2. Novelty**
>
> We respectfully but firmly disagree with the characterization of our work as "incremental." While our solution appears simple, this simplicity stems from our key novel insight (as listed below), while prior approaches to dynamic budget allocation in sparse attention (e.g., PyramidKV, AdaKV, DynamicKV) have overcomplicated the problem. Our core contributions are:
>
> - Algorithmic Insight: We demonstrate that the essential dynamism in sparse attention actually lies in the “distribution”, and can be achieved straightforwardly through top-p sampling, bypassing the need for complex, over-parameterized controllers prevalent in prior work.
> - Novel Hierarchical Architecture: We firstly introduce a hierarchical “selector-pruner” architecture, which successfully deploys top-p on existing top-k methods elegantly.
> System-Level Practicality: We show that top-p attention is not just theoretically sound but also hardware-friendly.
>
> As evidenced by our experiments, this approach outperforms existing dynamic methods while being more deployable. We contend that the simplicity achieved through fundamental understanding should not be conflated with incremental progress.
>
> **3. The “Contradiction” of the Performance**
>
> We appreciate that the reviewer raises an insightful question: why top-p is better than top-k with less tokens while also being worse than full attention? We are glad to provide a clarification.
>
> **Why top-p is worse than full attention?** The reason is that most *training-free* sparse attention designs are just numerical approximations to the full attention output, instead of actual *knowledge distillation/retrieve*. Training-free sparse attention is more like low-bit quantization, which inevitably causes accuracy loss. Trainable sparse attention (e.g. Native Sparse Attention from DeepSeek) has potential to surpass the full attention with better “focus”.
>
> **Why Twilight’s top-p is better than top-k with less tokens selected?** Twilight employs a hierarchical optimization process with two distinct budget parameters:
>
> - Initial Selection Budget (B0, follows the notation in Section 4.3): Our Token Selector (top-k) uses a conservative, relatively large B0 to maximize token coverage (Section 4.1). Crucially, this stage's computational cost is largely independent of B0's size so that we can use a conservative value of B0.
>
> - Pruned Budget (B1): The subsequent pruning stage aggressively reduces the token count to B1 using INT4-approximated importance scores.
>
> This two-stage approach provides key advantages over conventional top-k:
>
> - Broader Context Awareness: B0 > typical top-k budgets. Therefore Twilight sees more candidate tokens.
> - Precise Selection: INT4 scoring offers better importance estimation than typical top-k (which often reaches a sparsity of 1/16, as it is discussed in Section 4.
>
> This "look more, prune better" strategy consistently outperforms standard top-k attention in both accuracy and efficiency metrics. Note that there is also a trade-off in this two-stage method, which is calculated in Section 4.3.
>
> **4. Limited Long-Context Performance**
>
> We respectfully disagree with the characterization of Twilight as "inferior" to MagicPIG on long-context benchmarks. Our analysis of Tables 2 and 3 suggests the two methods are statistically comparable, with performance differences falling within expected experimental variance (<1%). Notably, Twilight with DS slightly outperforms MagicPIG in both Tables.
>
> We acknowledge MagicPIG as an excellent state-of-the-art sparse attention method that, like Twilight, demonstrates the advantages of moving beyond traditional top-k approaches. Both works provide independent validation of top-p sampling's superiority. Twilight has an advantage in respect of integration with existing algorithms due to its hierarchical architecture, while MagicPIG is a totally new algorithm-hardware co-design, with a unique sampling mechanism and attention computation workflow.
>
> **5. Average Post-Pruning Budget in Short-Context Evaluations**
>
> We appreciate the insightful advice from the reviewers. Below are the average post-pruning budget statistics in three short-context benchmarks. We will include them in our camera-ready version and we appreciate this opportunity to strengthen our experimental validation.
>
> | Benchmarks | Average Pruned Budget (Llama2) | Average Pruned Budget (Llama3) |
> |------|-------|-------|
> |GSM8K | 65.68 | 46.05 |
> | COQA | 71.01 | 41.09 |
> | PG-19 Perplexity | 70.18 | 72.91 |

---

> > ### Comment · Reviewer_mTkt · 2025-08-04
> >
> > Thanks for your detailed clarification. I will raise the scores.

---

> > > ### Author Response · Authors · 2025-08-04
> > >
> > > Thank you so much for your positive feedback and for deciding to raise the scores. I greatly appreciate your time and effort in reviewing my work and for considering my clarifications.

---

### Official Review · Reviewer_sLWS · 2025-07-06

**Clarity:** 3
**Significance:** 3
**Originality:** 3
**Rating:** 5
**Confidence:** 3

**Summary:**

The key idea of this paper is crystal clear: top-p is more effective than top-k for sparsity token selection. Although the key idea is quite simple, implementing top-p in state-of-the-art inference engines is not easy. The paper introduces a new method to select and prune tokens during the inference process and provides an efficient kernel implementation. The paper is well written and easy to follow.

**Questions:**

I expect more detailed in-depth analysis of the selected token distribution between top-p and top-k in experimental section.

It would be better to have more discussion on how to integrate it to existed frameworks other than Quest and FlashInfer.

**Ethical Concerns:**

["NO or VERY MINOR ethics concerns only"]

**Limitations:**

yes

**Quality:**

3

**Strengths And Weaknesses:**

The idea of this paper is simple, but unexpectedly effective. It also seems to be easily adopted by mainstream inference engines as well as most sparsity approaches.

The new method introduces a small amount of extra overhead during the selection and pruning phase. However, it is unclear whether this trade-off is worthwhile compared to the gains it provides (though 1.4x is significant in some cases).

---

> ### Author Rebuttal · Authors · 2025-07-31
>
> Thanks for your detailed review and insightful advice. Below, we address each concern and outline planned revisions to strengthen the paper.
>
> **1. Benefit-Overhead Tradeoff during Selection and Pruning**
>
> We appreciate the reviewer for mentioning the tradeoff between the extra overhead of Twilight and its benefits it brings. In Section 4.3, we discuss the overhead of Twilight and empirically analyze the time breakdown of Twilight’s attention operator in Figure 7, where the visualization makes this tradeoff particularly clear.
>
> **2. Analysis of Selected Token Distribution between Top-p and Top-k**
>
> We thank the reviewer for their insightful suggestion. In response, we will augment the experiment section with an in-depth analysis of token selection distributions (like Figure 3 in our paper, but more detailed) under both top-p and top-k sampling. This analysis will elucidate why top-p consistently achieves superior performance despite selecting fewer tokens, providing readers with a clearer understanding of its efficiency advantages.
>
> **3. Integration on Other Frameworks**
>
> We fully acknowledge the importance of system integration, especially given the practical focus of our method and its substantial system-level contributions. To clarify, Twilight's integration operates at multiple levels, which we categorize as follows:
>
> - **Kernel Library (FlashInfer)**. Currently Twilight leverages FlashInfer for high-performance implementations of key LLM operators (e.g., attention, FFN). While FlashInfer was chosen for its superior performance and flexibility, integration with alternative kernel libraries (e.g., Triton) is also possible. We plan to explore compiler-based approaches (e.g., Triton) in future work.
> - **Existing Sparse Algorithms (e.g., Quest)**. Here, "integration" refers to applying Twilight's optimizer to existing sparse algorithms. As detailed in Section 4, we demonstrate this through experiments with Quest, DS, and others.
> - **LLM Serving Frameworks (vLLM, SGLang)**. Serving frameworks typically build upon kernel libraries (e.g., FlashInfer is used by both vLLM and SGLang). Section 4.3 briefly discusses this integration layer, but we agree with the reviewer that concrete deployment examples would strengthen the work. We will add real-world implementation details in the camera-ready version.

---

### Comment · Area_Chair_g6sk · 2025-08-06

Dear Reviewers,

This is a gentle reminder that the discussion period will be concluded soon.

Please take the time to review the authors' responses (if provided) to your feedback and any questions you raised. Your careful consideration of their rebuttals is crucial for ensuring a fair and comprehensive evaluation of the submissions.

Following this, we kindly ask you to actively participate in the discussion to share your updated perspectives or align with fellow reviewers' insights as needed.

Finally, please ensure that you complete the review confirmation process by the specified deadline to finalize your assessment. This step is essential for moving forward with the decision-making process.

Thank you for your dedication and timely contributions to the NeurIPS review process.

Best regards,

The AC

---

### Decision · Program_Chairs · 2025-09-17

**Decision:**

Accept (spotlight)

**Comment:**

Twilight presents a novel framework for adaptive sparse attention in large language models (LLMs), leveraging top-p sampling to address static budget limitations in existing sparse attention methods. Its core innovation lies in a hierarchical "selector-pruner" architecture: a conservative base selector first identifies candidate tokens, followed by top-p pruning with INT4 quantization to dynamically adjust token budgets based on cumulative attention weights. This design enables aggressive pruning (up to 98% tokens) with minimal accuracy loss and 1.4x speedups over state-of-the-art methods, making it practical for long-context LLM inference.

Reviewers highlighted strengths in its simplicity, effectiveness, and real-world applicability. While some initially noted concerns—including overhead-benefit tradeoffs, incremental novelty perceptions, and INT4 cache overhead—the authors’ rebuttal comprehensively addressed these. They clarified pruning-stage costs are offset by reduced computation in long contexts, added analyses of top-p vs. top-k token distributions, detailed integration with frameworks like FlashInfer and VLLM, and distinguished Twilight from prefill-stage optimizations by focusing on decode-stage efficiency.

Most reviewers maintained or raised scores post-rebuttal, acknowledging the work’s significance. The framework’s compatibility with existing systems and empirical robustness across benchmarks (LongBench, GSM8K) further solidify its value.